# Impact of Image Context on Deep Learning for Classification of Teeth on Radiographs

**DOI:** 10.3390/jcm10081635

**Published:** 2021-04-12

**Authors:** Joachim Krois, Lisa Schneider, Falk Schwendicke

**Affiliations:** Department of Oral Diagnostics, Digital Health and Health Services Research, Charité-Universitätsmedizin, 14197 Berlin, Germany; Joachim.krois@charite.de (J.K.); lisa.schneider2@charite.de (L.S.)

**Keywords:** artificial intelligence, diagnostics, digital imaging/radiology, mathematical modeling

## Abstract

*Objectives:* We aimed to assess the impact of image context information on the accuracy of deep learning models for tooth classification on panoramic dental radiographs. *Methods:* Our dataset contained 5008 panoramic radiographs with a mean number of 25.2 teeth per image. Teeth were segmented bounding-box-wise and classified by one expert; this was validated by another expert. Tooth segments were cropped allowing for different context; the baseline size was 100% of each box and was scaled up to capture 150%, 200%, 250% and 300% to increase context. On each of the five generated datasets, ResNet-34 classification models were trained using the Adam optimizer with a learning rate of 0.001 over 25 epochs with a batch size of 16. A total of 20% of the data was used for testing; in subgroup analyses, models were tested only on specific tooth types. Feature visualization using gradient-weighted class activation mapping (Grad-CAM) was employed to visualize salient areas. *Results:* F1-scores increased monotonically from 0.77 in the base-case (100%) to 0.93 on the largest segments (300%; *p* = 0.0083; Mann–Kendall-test). Gains in accuracy were limited between 200% and 300%. This behavior was found for all tooth types except canines, where accuracy was much higher even for smaller segments and increasing context yielded only minimal gains. With increasing context salient areas were more widely distributed over each segment; at maximum segment size, the models assessed minimum 3–4 teeth as well as the interdental or inter-arch space to come to a classification. *Conclusions:* Context matters; classification accuracy increased significantly with increasing context.

## 1. Introduction

Deep learning (DL) is increasingly used for medical image analysis, and also in dentistry. Convolutional neural networks (CNNs) are a popular type of a neural network architecture that uses convolutions to extract meaningful features, such as edges, textures and other patterns from images. CNNs and variants thereof are commonly used to detect, segment and classify anatomic structures (hard or soft tissue landmarks, teeth) or pathologies (caries, periodontal bone loss, apical lesions, among others) [1]. For tooth classification, for example, models with sensitivities and specificities around 95–98% have been developed [2,3]. CNN learn from pairs of imagery data and labels (e.g., image labels, boxes encapsulating objects of interest, pixel masks) in a supervised way, eventually establishing a statistically based mapping of the input image to the output label. Once such a relationship is found the CNN can be applied on unseen images to classify them, detect objects on them or segment structures of interest in a pixel-wise manner [4].

One common feature of interest in dental computer vision is the tooth as the unit of learning. Most dental images contain more than one tooth; in case of peri-apical radiographs, 2–3 teeth; for bitewings 4–10 teeth, for panoramics up to 32 teeth, and for intraoral photography various numbers depending on magnification etc. In many instances, CNNs are trained not on the overall image, but a cropped one, centered around a particular tooth [5,6]. This increases the number of training units, while care is needed when partitioning data to not cross-contaminate data from the same patient between splits. Moreover, so far, to the best of our knowledge the downsides and limitations of training on cropped data have not yet been systematically evaluated. One may assume that, for example, tooth classification models may show lower performance when trained on cropped images as they miss context (e.g., anatomic landmarks, structures, other teeth) allowing to identify the cropped tooth’s position and class. If this was the case, then researchers should aim to increase the number of images used for training instead of cropping images to artificially increase the sample size but potentially decrease the model performance.

In the present study, we aimed to apply deep CNNs to classify individual cropped teeth from dental panoramic radiographs. We hypothesize that the classification performance of CNNs is significantly improved by providing a larger scope of image context information around the region of interest.

## 2. Materials and Methods

### 2.1. Study Design

In the present study, labeled bounding boxes, each encapsulating one tooth, were used to generate image segments from panoramic radiographs with a varying amount of context information around the region of inaterest, the centrally located tooth. Residual CNNs with 34 layers (ResNet-34) were trained, validated and tested on data of varying extent and context information [7]. The Mann–Kendall (M-K) test was employed to evaluate the impact of context information on the classification performance [8]. Reporting of this study follows the STARD guideline [9] and the Checklist for Artificial Intelligence in Dental Research [10].

### 2.2. Performance Metrics

The classification performance of the models was quantified by their accuracy and F1-score. Accuracy captures the proportion of correct classifications over all predictions of the model, while the F1-score considers the harmonic mean of precision (also referred to as positive predictive value (PPV)) and recall (also referred to as specificity). Due to a slight imbalance within the datasets, the F1-score was chosen as a primary metric. Secondary metrics were the area under the receiver operating characteristic curve (AUC), sensitivity, specificity, PPV, and negative predictive value (NPV). The model performance was evaluated on a hold-out test set consisting of 20% of the data.

### 2.3. Sample Size

A sample of 5008 retrospectively collected data from routine care was available, hence, no formal sample size estimation was performed.

### 2.4. Dataset and Reference Test

Our dataset contained 5008 dental panoramic radiographs with a mean number of 25.21 teeth per image. The data was collected between 2016 and 2019 during routine care at Charité-Universitätsmedizin Berlin. The collection of data was ethically approved (EA4/080/18). Totals of 50.8% and 49% of the panoramics originated from males and females, respectively (gender information for 0.2% were missing). The age at the time of visits ranged from 9 to 96 years, with a mean of 51.8 years. The radiographs were generated by radiographic devices from Sirona Densply (Bensheim, Germany), mainly ORTHOPHOS XG3D and ORTHOPHOS SL with CCD sensors.

The panoramic images were labeled bounding-box wise by dental experts. In total more than 50 experts were involved, including dental students, experienced dentist and researchers. Notably, we observed that labelling teeth is not a complex task and can be achieved with high accuracies by dental students, too. Each tooth was labeled based on the FDI notation. Each image was labelled once and then all labels were doublechecked by a second independent expert [11]. Each annotator independently assessed each image under standardized conditions using an in-house custom-built annotation tool as described before [12]. Prior to the annotation, the examiners were advised on how to place bounding boxes, including rotation etc.

### 2.5. Data Preparation, Model and Training

We utilized bounding box annotations to crop panoramic images, resulting in 124,314 cropped image segments. Naturally, the datasets are imbalanced considering the occurrence of different teeth. For example, the prevalence was lowest at 1.57% for tooth 18, 1.6% (28), 1.85% (38), 1.87% (48), and highest at 3.73% (32), 3.87% (33), 3.71% (42), 3.85% (43).

Cropping was repeated five times, each time with varying regional context information: The baseline bounding box size (100%) was scaled up to capture 150%, 200%, 250% and 300% of the baseline bounding box. The original aspect ratio was kept constant (see Figure 1). Note that the tooth considered for the classification task was consistently located within the center of each image segment.

The generated datasets were used to train five ResNet-34 classification models with the same set of hyperparameters. ResNet-34 has 34 layers and consists of a stack of residual blocks of CNNs that function as a feature extractor and a fully connected classification head (Appendix A). The model outputs a score for each possible label for the tooth located in the center of each input image. The label with the highest score was considered as the model’s prediction. Training, validation and testing of each model was exclusively performed on the dataset with the dedicated image segment size. Figure 2 provides a summarizing overview of the workflow.

The model optimization was performed with the adaptive moment estimation (Adam) optimizer with a learning rate of 0.001 and a categorical cross-entropy loss computed on the one-hot encoded representation of labels. Each model was trained over 25 epochs with a batch size of 16. The hyperparameter search was performed in a manual manner, as we only aimed at a model comparison and not at a maximal precision of one neural network. Notably, no image augmentation was used in the experiments as maximizing the generalizability of the models was not the focus of the study. To speed up model convergence, we applied transfer learning by reusing feature extraction blocks from a pre-trained model on ImageNet [12].

The model training was performed on 74,855 tooth segments (60%) of available data, while the remaining 40% (tooth segments) were split into validation (24,643 tooth segments) and test set (24,816 tooth segments). The split was performed on panoramic level, i.e., segments from one panoramic were kept in the same split. This also meant that the class distribution in each split was similar. The validation set was used to evaluate the model performance during training, while the test set was considered to quantify the model performance after training.

In a sensitivity analysis the test data set was split into different tooth groups (incisors (29%), canines (14%), premolars (27%) and molars (30%)) and the model performance on them was reported. All models were implemented in TensorFlow 2.3 and trained on a NVIDIA Quadro RTX 8000.

### 2.6. Explainability

Feature visualization was employed to visualize areas within an image that were particularly relevant for the model’s classification decision. We used the gradient-weighted class activation mapping (Grad-CAM) for this purpose [13], specifically, the relative importance, which is based on the gradient of the prediction score for a class with respect to feature map activations of the last convolutional layer, to generate feature maps. After a weighted summation of each feature map subset, the resulting maps were rescaled to allow overlaying it with the raw image data, thereby visualizing salient areas.

### 2.7. Statistical Analysis

For hypothesis testing, the Mann–Kendall (M-K) test [8], a nonparametric test for monotonic trends was applied on the series of F1-scores resulting from the trained models. In this study, a monotonic upward trend was characterized by a consistent increase of the F1-score through the extending context of the image segments. A significance level of <0.05 was considered statistically significant.

## 3. Results

When testing the base-case model, classification accuracy was 0.77, as was the F1-score, with a sensitivity of 0.74 and specificity of 0.99. Increasing the context, i.e., upscaling the image segment, increased the accuracy, with the maximum segment size of 300% yielding an accuracy, a F1 score and a sensitivity of 0.93, at a continuously high specificity of 0.99 (Table 1, Table 2 and Table 3). Notably, the improvements flattened with increasing context and gains in accuracy were limited between 200% and 300%. The increase in F1-score was statistically significant (*p* = 0.0083; M-K test).

When subgrouping tooth types, it was apparent that for all tooth types except canines and molars, a similar behavior was observed; the F1-score increased monotonically i.e., consistently increased through the extending context of the image segments. The increase flattened from 200% onwards. For canines, accuracy was much higher even for smaller segments and only minimal gains in accuracy reached by increasing context (Figure 3).

When assessing salient areas relevant for the classification (Figure 4), we found that with increasing context, the focus was removed from one specific spot of the image (for small segments, this was usually one interdental space or a coronal area of the tooth) and spread broader; at maximum segment size, the models assessed minimum 3–4 teeth as well as the interdental or inter-arch space to come to a classification, obviously making use of the provided context, as hypothesized.

## 4. Discussion

Deep learning applications are rapidly entering dentistry; one main component for accurate deep learning models is high accuracy. To improve accuracy on limited available datasets, cropping dental imagery, for example radiographs, and thereby multiplying the number of statistical units, is common, making use of the multiplicity of teeth in each image. Especially on dental panoramics, this approach has been employed.

Notably, cropping imagery around teeth comes with significant information loss, mainly context but also information on possible association of the units (i.e., clustering correlation) [14]. In the present study, we evaluated the impact of this information loss on one particular task, tooth classification on panoramic radiographs. We hypothesized that with increasing tooth segment size and the resulting increase in context, classification accuracy increases significantly. We accept this hypothesis.

Our findings require a deeper discussion. First, our primary finding can be linked to the fundamental design of CNNs. Early layers of CNNs have the task to detect low-level features such as vertical or horizontal edges derived from changes in brightness within the image. The later layers of CNNs combine these low-level features to detect high-level shapes or objects. Naturally, images with a larger scope of image context provide more low-level features that carry rich information of the image content. Consequently, the model has more possibilities to detect high-level objects, resulting in higher accuracy. Our experiments on explainability confirmed this; the models indeed used the available context in larger segments to come to their classification decisions.

Second, the resulting gains in accuracy were significant; we observed a 20% increase in F1-score when enlarging the segment size. Notably, the gain was mainly realized in the first enlargement steps; increases beyond 200% segment size yielded only limited accuracy gains, i.e., the benefit of context saturated. For researchers, this means that segmentation as a strategy remains useful, but should not be performed on tooth level, but allowing 2–3 teeth to remain present on an image, thereby balancing the positive effects of cropping and context.

Third, the tooth type played a relevant role for our particular task, tooth classification. While all teeth except canines showed the described behavior, benefitting from context, the classification of canines showed already high accuracy on the baseline image segments and accuracy did not increase with context. This is likely as only one canine can be present per segment regardless of the segment size given the availability of only one canine per quadrant. In this case, misclassification is generally less likely. This highlights one other finding: Based on our findings it seems that models were able to identify each tooth type reliably based on their specific anatomic features regardless of the segment size (e.g., canines seem clearly distinguishable from other tooth types), but that the positioning of the specific tooth type and hence the subclassification (e.g., first versus second molar) required context (which for canines, played no role, as no discrimination from other canines was needed.)

Fourth, the age range of individuals in our sample was large–and with it the number of teeth (and the teeth present, as outlined in the prevalence data). This was needed to allow generalizability of the underlying tooth classification models. These should work on fully dentate or partially dentate radiographs, i.e., different age ranges. Finally, very similar model performance metrics on the validation and test set confirmed that the model was not prone to overfitting. This study comes with a number of limitations. First, we employed only one specific model architecture, ResNet-34, for one specific deep learning task, tooth classification on panoramics, on data from one center. We hence do not claim generalizability of our models’ performance as well as our findings. It may be that other models show different behavior, and it is likely that the effect of context on other tasks will differ. Second, we did not include any image augmentation methods, since the focus of this work was on the impact of varying input data on model performance and not on maximizing the generalizability of the trained models. The rotation before training assured more standardized input image data, as the tooth orientation may be highly variable, depending on the dental status of the subject (Figure 1).

Furthermore, we did not perform any actions against the class imbalance of the dataset. As a result, our subgroup analyses on different tooth groups are based on varying numbers of images per group, which limits the comparability of accuracies in different groups. It is however unlikely that the direction of the observed effects will be affected, but possibly their magnitude. Moreover, implants and restored teeth were included in this tooth classification task. Implants or crowned teeth provide less distinguishable features than natural teeth and thereby may affect the performance magnitude of the models. Also, the distribution of implants and crowns in-between different tooth groups may differ and affect the model performance. The decision to include implants and restored teeth was made given the larger scope of this study; the general detection and classification of dental units on panoramic radiographs to later on allow associating dental restorations or pathologies with specific teeth. To achieve this, CNNs must be exposed to the feature space of healthy and restored teeth and implants during training, irrespective of whether the model is trained on full-sized or cropped images. A suggestion for the size of cropped images used for training should therefore be based on an evaluation of healthy and restored teeth and implants. Admittedly, classifying implants is harder given that implant class is distinguishable from implant position, not their anatomy. Note, that the prevalence of implants was low (3.2%), though, and the bias stemming from this aspect should be limited.

Further studies are needed to explore these effects in more detail.

## 5. Conclusions

Context matters; classification accuracy increased significantly with increasing context. Cropping imagery during training may hamper deep learning accuracy. Our findings suggest training models on larger cropped image segments with 2–3 teeth to balance the positive effects of cropping and context.

## Figures and Tables

**Figure 1 jcm-10-01635-f001:**
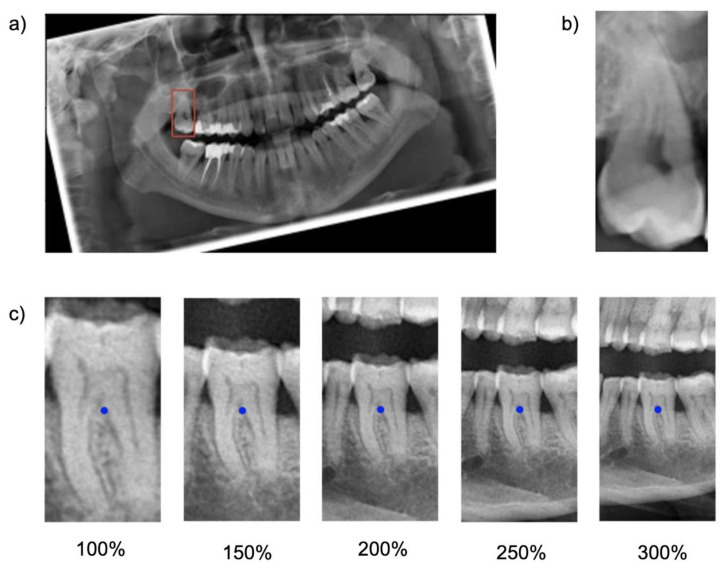
Visualization of data preparation steps. (**a**) Panoramic radiograph with an annotated bounding box around one tooth. The image is rotated as required by the cropping algorithm, which extracts the image segment captured by the bounding box. (**b**) Baseline size (100%) of the image segment covering the area captured of the bounding box. (**c**) Visualization of increasing image segment sizes from left to right. 100% captured image segments with the size of the annotation bounding box. 300% covered image segments with the size of the bounding box upscaled by a factor of 3.

**Figure 2 jcm-10-01635-f002:**
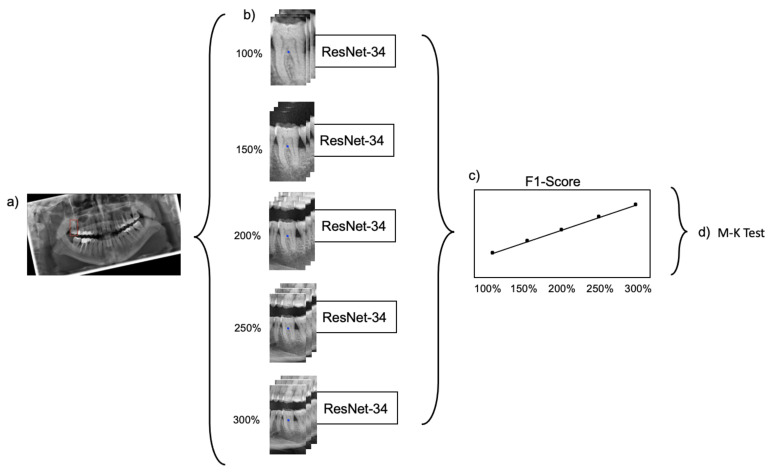
Visualization of workflow. (**a**) Panoramic radiograph with an annotated bounding box around one tooth. (**b**) Extraction of image segments with varying image context information. Training of one ResNet-34 model for each crop size. (**c**) Collection of the F1-scores based on the performance on the unseen test set. (**d**) Application of the Mann–Kenndall test.

**Figure 3 jcm-10-01635-f003:**
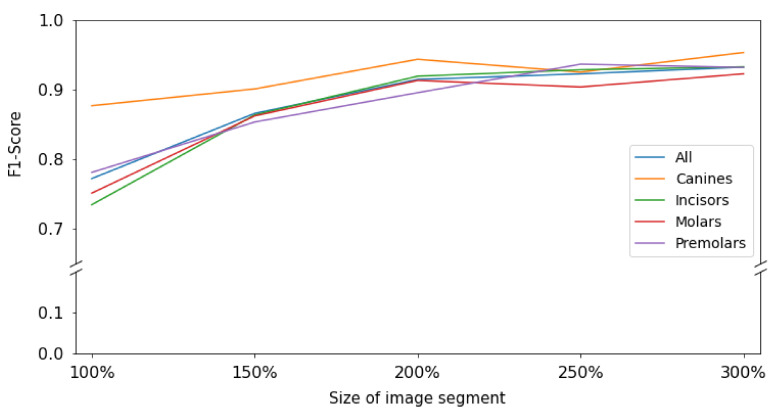
Visualization of F1-score for the test set for each image context size grouped per tooth group and over all tooth groups. The baseline size of 100% captured image segments with the size of the annotation bounding box; the image segment size of 300% covered image segments with the size of the bounding box upscaled by a factor of 3.

**Figure 4 jcm-10-01635-f004:**
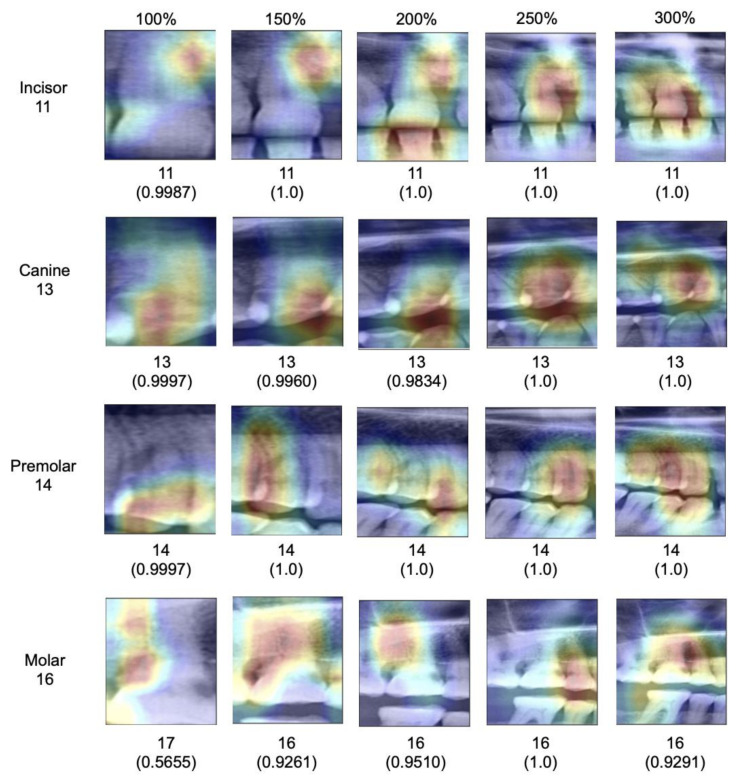
Visualization of contributing feature maps for random teeth from the test set. Images of incisors, canines, premolars and molars (lines) and different crop sizes (columns) are presented. The original images and the salient areas most relevant for the model’s decision (highlighted in yellow to red) are shown. Ground truth labels are shown on the left. Predicted labels and confidence scores (values closer to 1 indicate higher confidence in the model’s classification) are noted below.

**Table 1 jcm-10-01635-t001:** Mean performance of the models on hold-out test set. Performances of train and validation set are reported in Table 2 and Table 3.

Crop Size	Accuracy	F1-Score	AUC	Sensitivity	Specificity	PPV	NPV
100%	0.769	0.773	0.984	0.740	0.994	0.81	0.992
150%	0.862	0.866	0.993	0.846	0.997	0.888	0.995
200%	0.913	0.915	0.996	0.903	0.998	0.928	0.997
250%	0.920	0.923	0.995	0.914	0.998	0.932	0.997
300%	0.931	0.933	0.994	0.927	0.998	0.939	0.998

AUC area-under-the-curve. PPV/NPV positive/negative predictive value.

**Table 2 jcm-10-01635-t002:** Mean performance of the models on train set.

Crop Size	Accuracy	F1-Score	AUC	Sensitivity	Specificity	PPV	NPV
100%	0.826	0.827	0.992	0.798	0.996	0.860	0.993
150%	0.911	0.912	0.997	0.898	0.998	0.929	0.997
200%	0.958	0.958	0.999	0.951	0.999	0.965	0.998
250%	0.957	0.958	0.999	0.952	0.999	0.964	0.998
300%	0.968	0.968	0.999	0.965	0.999	0.971	0.999

AUC area-under-the-curve. PPV/NPV positive/negative predictive value.

**Table 3 jcm-10-01635-t003:** Mean performance of the models on validation set.

Crop Size	Accuracy	F1-Score	AUC	Sensitivity	Specificity	PPV	NPV
100%	0.767	0.720	0.984	0.738	0.994	0.811	0.992
150%	0.870	0.873	0.993	0.854	0.997	0.894	0.995
200%	0.916	0.918	0.995	0.907	0.998	0.931	0.997
250%	0.923	0.924	0.994	0.916	0.998	0.933	0.997
300%	0.933	0.934	0.993	0.928	0.998	0.940	0.998

AUC area-under-the-curve. PPV/NPV positive/negative predictive value.

## Data Availability

The data presented in this study may be available on request from the corresponding author within data protection boundaries.

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
