# Peer review of "Impact of Image Context on Deep Learning for Classification of Teeth on Radiographs"

_jcm, 2021, doi:10.3390/jcm10081635_

Round 1

Reviewer 1 Report

Thank you for the opportunity to review this manuscript.

The authors developed an AI-based automated classification model for identifying each tooth from panoramic radiographs.

The results of the study indicate that context matters, and that classification accuracy increased significantly with increasing context, which is very useful for understanding the nature of classification model, ResNet-34. However, from my understanding, the classification model only classifies the input image into the trained labels. It does not indicate the region of interest, which in this case is each tooth to be identified. In order to apply this AI model to clinical practice, the user should crop the image of each tooth. The authors should have used faster R-CNN or detection models such as YOLO and SSD. These models can detect the object of interest, i.e. each tooth, then classifies them.

Why were the images rotated before cropping? If augmentation such as rotation were performed, then it would have had a better performance and not require rotation of the radiograph prior to cropping.  

One major concern is the inclusion of dental implants. Why did the authors include dental implants? Authors had enough number of cropped tooth images. If the dental implants were included, they should have been labeled independently. Labeling them as the natural teeth would lead to a significant increase in error, as their shape is totally different from natural teeth. The tooth number from observing a dental implant is almost impossible by only observing the implant. How many dental implants were included? Can you show the accuracy measures of the dental implants separately? I suggest retraining the model without the dental implants, because the results of the study can be misleading.

Minor comments

Please indicate the type of sensor for the panoramic machines. (digital ccd or CR)

Please indicate if the experts who labeled the images are dentists. Provide their initials, and the year of clinical experience. If the experts are not dentists, it should be stated in the manuscript.

Why is there such a low prevalence of third molars? I believe it is much lower than the natural occurence.

What does it mean that the accuracy increased “monotonically”?

What is the possible explanation for the high accuracy for the canines?

In Fig 4, please show the prediction values for each image.

Author Response

The results of the study indicate that context matters, and that classification accuracy increased significantly with increasing context, which is very useful for understanding the nature of classification model, ResNet-34. However, from my understanding, the classification model only classifies the input image into the trained labels. It does not indicate the region of interest, which in this case is each tooth to be identified. In order to apply this AI model to clinical practice, the user should crop the image of each tooth. The authors should have used faster R-CNN or detection models such as YOLO and SSD. These models can detect the object of interest, i.e. each tooth, then classifies them.

Our response: The author is correct that an end-to-end modeling approach using e.g. YOLO or other type of object detections models may be used for classification (e.g. tooth number) and position on the radiograph. However, using object detection models would not allow to investigate the research question that we were interested in. Our research questions was inspired by the fact that numerous publications in this field used cropped dental radiographs and applied various different models on them (including but not limited to tooth number classification). Hence, we used a model on crops and evaluated the impact of additional context on image classification and not object detection.

Why were the images rotated before cropping? If augmentation such as rotation were performed, then it would have had a better performance and not require rotation of the radiograph prior to cropping.  

Our response: No data augmentation and hence no random rotations were performed during training.

One major concern is the inclusion of dental implants. Why did the authors include dental implants? Authors had enough number of cropped tooth images. If the dental implants were included, they should have been labeled independently. Labeling them as the natural teeth would lead to a significant increase in error, as their shape is totally different from natural teeth. The tooth number from observing a dental implant is almost impossible by only observing the implant. How many dental implants were included? Can you show the accuracy measures of the dental implants separately? I suggest retraining the model without the dental implants, because the results of the study can be misleading.

Our response: This study emanates as work on the general  detection and classification of dental units on panoramic radiographs. In such work, for example embedded in clinically useful software where dental restorations or pathologies are then associated with specific teeth, it is (a) common to meet implants and (b) useful to assign a class (position) to implants too. Admittedly, this task is harder given that implants are only distinguishable from their position, not their anatomy. We highlight this point now. Note, that the prevalence of implants was low, though, and the bias stemming from this aspect should be limited.

Please indicate the type of sensor for the panoramic machines. (digital ccd or CR)

Our response: This was added.

Please indicate if the experts who labeled the images are dentists. Provide their initials, and the year of clinical experience. If the experts are not dentists, it should be stated in the manuscript.

Our response: Tooth labeling was done in a repeated fashion. At least 2 dental experts confirmed the tooth label. In total more than 50 experts have been involved, including dental students, experienced dentist and researchers. Further, we observed that labelling teeth is not a very complex task and can be achieved with high accuracies even by dental students. Naming all of them as co-authors is not feasible.

Why is there such a low prevalence of third molars? I believe it is much lower than the natural occurrence.

Our response: While we cannot ascertain the reasons for this, we highlight it. It is possible that given this age range, many patients experienced removal of third molars in earlier age, resulting in lower prevalence than what has been reported on younger populations.

What does it mean that the accuracy increased “monotonically”?

Our response: With every step, accuracy increased, i.e. the increase was monotone. We deleted this.

What is the possible explanation for the high accuracy for the canines?

Our response: This has been discussed – it is likely due to the fact that misclassification of canines is less likely per image as only one canine can be present per cropped image.

In Fig 4, please show the prediction values for each image.

Our response: This was done.

Reviewer 2 Report

This is a well-written and interesting manuscript aimed at applying deep CNNs to classify individual teeth from dental panoramic radiographs. I have a few comments:
1) Please provide clarification regarding the issues surrounding the large age distribution of 9 to 96 years where teeth could significantly vary
2) Please clarify in the methodology the tuning parameters used to reach the final model (i.e. batch size, epoch selection).
3) I recommend a flowchart of the statistical methodology to help the reader understand the stages involved in reaching the final model
4) Please clarify if any adjustments for multiple testing to reduce the risk of a Type I error were performed
5) Please clarify which data set the results in Table 1 and Figure 2 related to (e.g. testing / validation data?). Please provide the full set of results.
6) Please clarify if overfitting was tested and accounted for

Author Response

1) Please provide clarification regarding the issues surrounding the large age distribution of 9 to 96 years where teeth could significantly vary

Our response: Indeed, the age range was large – and with it the number of teeth (and the teeth present, as outlined in the prevalence data presented). As explained for reviewer 1, this study emanates as work on the general  detection and classification of dental units on panoramic radiographs, which would allow to associated dental restorations or pathologies with specific teeth. Of course, to have some kind of generalizability, the underlying tooth classification should work on fully dentate or partially dentate radiographs, i.e. different age ranges. We add that now.

2) Please clarify in the methodology the tuning parameters used to reach the final model (i.e. batch size, epoch selection).

Our response: This was added.

3) I recommend a flowchart of the statistical methodology to help the reader understand the stages involved in reaching the final model.

Our response: This was added.

4) Please clarify if any adjustments for multiple testing to reduce the risk of a Type I error were performed

Our response: No adjustments were undertaken, as the Mann-Kendall-testwas only applied once on the trend in F1-score.

5) Please clarify which data set the results in Table 1 and Figure 2 related to (e.g. testing / validation data?). Please provide the full set of results.

Our response: This was added.

6) Please clarify if overfitting was tested and accounted for

Our response: We expanded on this based on the data we show for the validation and test dataset performance.

Round 2

Reviewer 2 Report

Thank you for your comments and responses to my previous suggestions and questions.